# Effects of Mo_2_C on Microstructures and Comprehensive Properties of Ti(C, N)-Based Cermets Prepared Using Spark Plasma Sintering

**DOI:** 10.3390/molecules30030492

**Published:** 2025-01-23

**Authors:** Mu Qiao, Zhiwei Zhao, Guoguo Zhang, Hongjuan Zheng

**Affiliations:** College of Materials Science and Engineering, Henan University of Technology, Zhengzhou 450001, China; 18736386359@163.com (M.Q.); zgg17335220016@163.com (G.Z.); hongjuan_zheng@haut.edu.cn (H.Z.)

**Keywords:** Ti(C, N), spark plasma sintering, molybdenum carbide, material properties

## Abstract

Ti(C, N)-based cermets are problematic in practical production applications due to their brittleness. To improve this defect, Ti(C, N)-based cermets were prepared under different sintering environments using a spark plasma sintering (SPS) device with different contents of Mo_2_C and 25 wt.% of nano tungsten carbide as additives. By means of microstructural analysis and comprehensive performance tests, the Ti(C, N) cermet with 6 wt.% Mo_2_C content showed the best comprehensive performance when sintered at 1450 °C under a pressure of 25 MPa with a holding time of 16 min. The density of this metal-ceramic was 6.27 g/cm^3^, the Vickers hardness was HV 2731, and the fracture toughness was 10.1 MPa·m^1/2^, which increased the density by 15%, the hardness by 63%, and the fracture toughness by 84% compared with the ceramic without added Mo_2_C. Densification of cermets can be promoted using SPS. The moderate addition of Mo_2_C can improve the wettability between the bonded phase and the hard phase, and its joint action with tungsten carbide can promote the formation of a ring structure and inhibit the growth of core-structure grains to enhance the toughness of the ceramic.

## 1. Introduction

Cutting tools are usually made of materials with good mechanical properties, such as ceramic-metal composites with high hardness and good fracture toughness. They consist mainly of WC and Co and are known as cemented carbides [1,2,3]; however, the yearly reduction in the supply of W and Co has increased its manufacturing cost and made it more difficult to meet the demand for machining of new difficult-to-machine materials [4,5,6]. Therefore, it is necessary to find other materials that have similar mechanical properties to cemented carbide to replace it. Ti(C, N)-based cermet, as one of the alternatives to WC-based cemented carbides and also with good mechanical properties as cemented carbides, can be the main targets of current research. Ti(C, N)-based cermets stand out due to their remarkable attributes, including high thermal hardness, strong resistance to wear, a low coefficient of friction, and exceptional chemical stability [7,8,9]. In addition, global reserves of Ti are higher than those of W, resulting in a lower price for Ti than for W, which makes Ti(C, N)-based metal-ceramic tools cheaper to manufacture than WC-based cemented carbide tools [10,11,12].

Ti(C, N)-based cermets are a composite material incorporating ceramic hard phases (Ti(C, N)) and metal binder phases (Co, Ni), providing both malleable toughness inherent to metals and wear resistant properties typical of ceramics [13,14,15]. However, Ti(C, N)-based cermets encounter significant challenges in practical applications due to their low fracture toughness and poor wettability between the binder phase and the Ti(C, N) hard phases. To enhance the mechanical properties of Ti(C, N)-based cermets, a common strategy is to introduce additional carbides. The formation of a distinctive core-ring structure in titanium carbonitride is facilitated by the incorporation of carbides. In this structure, the core consists mainly of a dark hard phase of titanium carbonitride, whereas the ring is composed of a gray rim brittle phase of (Ti, M) (C, N), where M denotes alloy constituents [16,17,18]. The core-ring structure enhances wettability between the hard and bonding phases, and the encapsulation of the hard phase by the ring phase effectively inhibits grain growth, thereby enhancing the overall mechanical performance of the cermet [19,20,21]. Previous studies show that the incorporation of Mo_2_C into metal-ceramic composites resulted in enhanced wetting properties between the binder and the hard phase of Ti(C, N)-based cermets. Additionally, this addition suppresses the growth of Ti(C, N) particles as liquid phase sintering (LPS) proceeds [22,23,24]. Ultimately, adding additives gives Ti(C, N)-based cermets a more homogeneous ring structure and a more homogeneous, finer core structure: these changes lead to better overall performance of the cermets. In addition, the addition of WC promotes the generation of edge phases, while concurrently improving Ti(C, N) hard phases in terms of their lattice parameters [25,26]. This improvement curtails abnormal grain growth, thereby reducing the grain size and fortifying Ti(C, N).

The previous method is only one of the ways of improving the properties of Ti(C, N)-based cermets, the properties of which can also be enhanced by means of different sintering methods. Spark plasma sintering (SPS) equipment is presently employed in numerous research endeavors directed towards cermet fabrication, primarily due to its ability to achieve rapid sintering, low-temperature processing, and uniform sintering of samples [27,28,29]. Currently, little research is available about the manufacturing of Ti(C, N)-based cermets that incorporate Mo_2_C and nano WC through SPS. Ti(C, N)-based cermets were prepared with Mo_2_C and nano WC as additives in the current research using SPS under four conditions. Combined with microstructural analysis and comprehensive performance tests, the most rational method of preparation was explored.

## 2. Results and Discussion

### 2.1. Phase Compositions and Binding States

XRD patterns of Ti(C, N)-based cermet samples with added molybdenum carbide and prepared using SPS are illustrated in Figure 1.

Figure 1a shows XRD patterns of Ti(C, N) samples treated with various amounts of molybdenum carbide, subjected to the identical SPS process (sintering at 1450 °C, under 30 MPa, and holding for 16 min). The XRD patterns are primarily composed of diffraction peaks of WC and Ti(C, N). No diffraction peaks of Mo_2_C were observed, possibly because of the smaller ionic radius of Mo^4+^ compared with W^4+^, making it easier for Mo^4+^ to diffuse into Ti(C, N)-based hard phase [30]. Finally, molybdenum carbide exhibits good solid-solution effects with the hard phase, and Mo_2_C has already dissolved, thus forming a solid solution of (Ti, M) (C, N). When different amounts of Mo_2_C are added, the WC diffraction peaks begin to show a lower intensity, indicative of certain impairment of Mo_2_C content on the relative content and phase composition of the Ti(C, N)-based cermet. During the sintering process, WC can also form a solid solution of (Ti, M) (C, N) by dissolving with other carbides and can also dissolve with Co. An excess of WC can affect the strength of the sample. Under an Mo_2_C content of 6 wt.%, the WC diffraction peaks have relatively low intensity. With an increase in the Mo_2_C content from 2 wt.% to 8 wt.%, diffraction peaks of the Ti(C, N) phase change to a lower angle. This indicates an increase in the lattice spacing of the Ti(C, N) phase according to Bragg’s equation (2*d* sin*θ* = *kλ*).

As the sintering temperature varies, the following solid solution reactions may occur in metal-ceramics:Mo_2_C + Ti(C, N) → (Mo, Ti) (C, N) + Ti(C, N)(1)WC + Ti(C, N) → (Ti, W) (C, N) + Ti(C, N)(2)Mo_2_C + WC + Ti(C, N) → (Ti, W, Mo) (C, N) + Ti(C, N)(3)(Mo, Ti) (C, N) + (Ti, W) (C, N) + Ti(C, N) → (Ti, Mo, W,) (C, N)(4)

In the early stage of sintering, the raw material particles are uniformly distributed, albeit with only limited diffusion [31]. Between 1000 °C and 1250 °C, Ti(C, N) particles and carbides (Mo_2_C and WC) begin to partially dissolve into the bonding phase (Co), as illustrated in reactions (1) and (2). The dissolution of carbide inhibits the excessive dissolution of Ti(C, N), and undissolved Ti(C, N) begins to form a black core phase. Meanwhile, the dissolved Ti, C, and N from Ti(C, N) particles diffuse into the metal-bonded phase and combine with dissolved Mo and W to form an initial inner rim structure on the particle surface. In the liquid-phase sintering stage (after 1250 °C), the reactions are primarily governed by reaction Equations (3) and (4). The complete melting of the liquid phase promotes the dissolution–precipitation processes of Ti(C, N) and carbides (Mo_2_C and WC). At this stage, Mo_2_C is completely dissolved, and as the holding time increases, WC also becomes fully dissolved. Subsequently, Mo and W from these dissolved carbides, together with dissolved Ti(C, N), precipitate on the surface of Ti(C, N) particles, forming a continuous and complete (Ti, Mo, W) (C, N) rim structure.

Figure 1b illustrates the XRD patterns of samples with a consistent molybdenum carbide addition (x = 6.0 wt.%), holding time (16 min), sintering pressure (30 MPa), and varying sintering temperatures (1350–1500 °C). As shown in Figure 1b, increasing the sintering temperature leads to a decrease in the diffraction peaks of Ti(C, N) and WC, indicating that higher temperatures enhance their dissolution–precipitation process. The accelerated dissolution–precipitation process rapidly forms the rim phase (Ti, Mo, W) (C, N) from dissolved Ti(C, N), Mo, and W. On the one hand, the quick emergence of this rim phase fosters the development of small, stable cores of undissolved Ti(C, N). On the other hand, excessively high temperatures can drive the formation of a thicker rim phase through an overly rapid dissolution–precipitation process.

Figure 1c illustrates the XRD patterns of samples with a consistent molybdenum carbide addition (x = 6.0 wt.%), sintering temperature (1450 °C), sintering pressure (30 MPa), and varying holding times (8 min–20 min). As shown in Figure 1c, a shorter holding time (8 min) produced higher Ti(C, N) and WC peaks, indicating less dissolution of these phases. Consequently, this may result in a larger core phase and a thinner, nonuniform rim phase. When the holding time was extended to 12 min, the diffraction peak of WC disappeared, and the diffraction peak of Ti(C, N) shifted to higher angles. This suggests that the longer holding time could enhance the dissolution–precipitation process of carbides, allowing more Mo and W ions to diffuse into the Ti(C, N) lattice. At a holding time of 20 min, the diffraction peak of Ti(C, N) shifts to a significantly higher angle, potentially causing the rim phase to become excessively thick. Furthermore, an overly long holding time may also result in anomalous growth of the core phase.

Figure 1d illustrates the XRD patterns of samples with a consistent molybdenum carbide addition (x = 6.0 wt.%), sintering temperature (1450 °C), holding time (16 min), and varying sintering pressures (20 MPa–35 MPa). Increasing the sintering pressure enhances the rates of dissolution and precipitation. As shown in Figure 1d, the diffraction peak of Ti(C, N) shifts to higher angles with increasing pressure. This indicates that more Mo_2_C and WC are incorporated into the Ti(C, N) lattice, promoting the formation of the (Ti, Mo, W) (C, N) phase and resulting in an increased thickness of the rim phase. However, when the pressure is increased to 30 MPa, the diffraction peaks of Ti(C, N) shift to lower angles. This shift may be because the precipitation rate exceeds the dissolution rate under high pressure, leading to lattice expansion in the rim phase. The diffraction peak intensity of Ti(C, N) is highest when the pressure increases to 30 MPa, indicating that the material may experience grain coarsening. Additionally, excessive sintering pressure can lead to the thickening of the rim phase and the spillage of the liquid phase during the sintering process, thereby deteriorating the material properties.

To explore the composition of chemical elements and bonding state, X-ray photoelectron spectroscopy (XPS) was applied to measure ceramic samples with a molybdenum carbide content of 6 wt.% and sintering conditions of 1450 °C, 16 min, and 30 MPa. As Figure 2a shows, titanium, tungsten, cobalt, carbon, nitrogen, molybdenum, and oxygen are the main constituents of the surface of the ceramic sample. A total of four peaks (A, B, C, and D) are found on the XPS spectrum in the C1s energy region of the ceramic sample, corresponding to peak binding energies separately of 284.7 eV, 288.05 eV, 285.8 eV, and 280.5 eV (Figure 2b). Peak A corresponds to the graphite on the surface of the carbide and some free carbon [32]. Peak B corresponds to other carbides of titanium and tungsten. The C-N bond and TiC correspond to peaks C and D, respectively. Figure 2c shows that the three peaks (A, B, and C) each have binding energies of 399.6 eV, 396.5 eV, and 402.5 eV. The N-C bond corresponds to peak A, TiN corresponds to peak B, and graphitized nitrogen corresponds to peak C.

Likewise, the XPS analysis of this cermet sample implies that four peaks, namely, A, B, C, and D, are present in the Ti2p energy region (Figure 2d). The binding energies corresponding to these four peaks A, B, C, and D are 464 eV, 458.35 eV, 460.30 eV, and 454.55 eV, respectively. Peak A corresponds to other compounds of titanium. TiO_2_, TiC, and TiN associated with Ti2p correspond to the peaks of the binding energies B, C, and D, respectively [33,34]. As shown in Figure 2e, the five peaks correspond to peak energies of 35.45 eV (peak A), 37.65 eV (peak B), 37.05 eV (peak C), 33.85 eV (peak D), and 31.8 eV (peak E), respectively. WO_3_, tungsten and tungsten carbide, which belong to W2p, correspond to peaks A, D, and E, respectively [35,36]. The other compounds of tungsten correspond to the remaining peaks B and C.

Figure 2f reveals the presence of two distinct peaks (peaks A and B) on the XPS spectra of the sample in the O1s energy region, corresponding to binding energies of 532.15 eV and 530.25 eV, respectively. Peak A corresponds to OH^−^, while peak B is probably WO_3_ or TiO_2_ [37]. As displayed in Figure 2g, there are a total of four peaks associated with binding energies, and their corresponding binding energies are 781.7 eV (peak A), 786.9 eV (peak B), 796.7 eV (peak C), and 802.3 eV (peak D). Among these peaks, only peak A corresponds to cobalt oxide [38], while the other three peaks correspond to other compounds of cobalt. XPS analysis of cermet samples unveils four distinctive peaks within the Mo3d energy region, as shown in Figure 2h. These peaks A, B, C, and D are associated with binding energies of 235.6 eV, 232.5 eV, 231.6 eV, and 234.0 eV, respectively. Peaks A and B correspond to the spin-orbit components of Mo3d5/2 and Mo3d3/2 of MoO_3_, respectively. Peak C belongs to single crystal MoO_3_ [39], while peak D is caused by Mo^4+^ in MoO_2_.

### 2.2. Microstructures

Surface morphologies of and pore distributions in the cermet samples were analyzed in Figure 3 to investigate the impact of molybdenum carbide content. All samples in Figure 3 were prepared at a sintering temperature of 1450 °C, a holding time of 16 min, and a sintering pressure of 30 MPa. This analysis was undertaken using SEM in backscatter mode and mapping observation.

As illustrated in Figure 3, the bright phase image is the WC phase, and the dark phase image is the Ti(C, N) phase, between which is the cobalt phase. Figure 3 demonstrates that the sample consists of two types of grains. The black and grey hard phases separately belong to the core and ring structures. Some Ti(C, N) particles form the black hard phase (Figure 3) due to incomplete dissolution during LPS, which becomes the core structure in the core-ring structure [40,41]. The grey hard phase represents a solid solution comprising Ti(C, N), tungsten carbide, and Mo_2_C, which is formed mainly by solid solution reaction. Different colors appear because the backscattering images are white for the higher average atomic number (W, Mo) and black for the lower average atomic number (Ti). The incorporation of Mo_2_C and tungsten carbide promotes the formation of a ring structure dominated by solid solution particles of (Ti, W, Mo) (C, N) during the sintering of Ti(C, N)-based cermets. The causes for the grey color of the ring structure in the SEM images are a lower average number of atoms and higher contents of molybdenum and tungsten [42]. A white bound phase can be observed close to the Ti(C, N) hard phases, combined with XRD analysis in Figure 1. The composition of the white-bound phase is found to be mainly cobalt. Based on previous studies and Gibbs free energy calculations, the stability of Mo_2_C is significantly lower than that of common carbides such as TaC and NbC [43]. Therefore, the solubility of Mo_2_C should be higher than that of other carbides, and a thicker annular phase can be obtained by using carbides such as Mo_2_C [44], resulting in better inhibition of the Ti(C, N) grain growth.

The average grain size of the black hardcore phase in Figure 3a–e is 0.97 μm, 0.88 μm, 0.71 μm, 0.64 μm, and 0.80 μm. According to Figure 3a–c, there are more porous regions in these three groups of samples. As the content of molybdenum carbide increases to 6 wt.%, the Ti(C, N) hard phase is distributed more uniformly and becomes finer, along with the reduction in porosity, as shown in Figure 3d. This could be a consequence of the higher carbide additive content, facilitating a more comprehensive diffusion of molybdenum atoms between the hard and binder phases during sintering. This thus improves the wettability of Ti(C, N) and cobalt [16,22]. By doing so, the Ti(C, N) hard phase is better combined with the Co binder phase, reducing porosity while increasing its compactness. At the same time, Figure 3c,d demonstrates that the grain size of the black core phase significantly decreases, while the thickness of the grey rim phase increases and becomes uniformly distributed with Mo_2_C additions of 4% and 6%. The analysis of the corresponding EDS patterns reveals that as Mo_2_C addition increases, the Ti distribution area gradually shrinks and the Ti content in the edge phase decreases, whereas the distribution of W and Mo in the edge phase increases significantly. This suggests that the core size diminishes with higher Mo_2_C content, while the rim phase thickens as a result of increased Mo_2_C addition. However, as illustrated in Figure 3e, the thickness and porosity of the ring phase have increased, possibly due to an excess of Mo_2_C additive. The excessive thickness of the ring structure and excessive porosity can adversely affect the material properties; therefore, the amount of molybdenum carbide as an additive should not be excessive [19].

Figure 4 presents SEM images illustrating the microstructure of 61Ti(C, N)-25WC-8Co-6Mo_2_C samples prepared under varying sintering temperatures. The average grain size of the black hardcore phase in Figure 4a–d is 0.75 μm, 0.68 μm, 0.58 μm, and 0.66 μm. As observed in Figure 4a, a lower sintering temperature (1350 °C) results in Ti(C, N) particles that have not yet been significantly dissolved, and the rim phase remains thin. Furthermore, the sample exhibits insufficient densification, characterized by a large number of pores, which detrimentally affects the overall material performance. Figure 4a,b demonstrates that, with increasing sintering temperature, the grain size of the black core phase decreases significantly, while the thickness of the grey rim phase increases and becomes uniformly distributed. An analysis of the corresponding EDS patterns reveals a gradual contraction of the Ti distribution region, a more homogeneous distribution of W and Mo, and a significant increase in the distribution of W and Mo within the edge phase. These observations indicate that higher sintering temperatures promote the formation of the (Ti, Mo, W)(C, N) rim phase, aligning with the conclusions drawn from Figure 1b. As shown in Figure 4c, the core-ring structure of this cermet sample becomes more complete, and the edges in front of the grey and black hard phases are clearer. Due to the more suitable sintering temperature (1450 °C), this sample also has a finer Ti(C, N) grain size, and the pores in the figure are barely visible. When the temperature is increased to 1500 °C, as illustrated in Figure 4d, the non-uniformity observed in the microstructure of the material can be ascribed to the differential dissolution rates of small and coarse particles during sintering. Specifically, finer particles exhibit a quicker dissolution rate, making them more susceptible to dissolution and subsequent deposition onto the coarser particles [45,46]; some grains also grow in an uneven manner, resulting in large irregular grains, due to the excessively thick ring phase.

Figure 5 presents SEM images illustrating the microstructure of 61Ti(C, N)-25WC-8Co-6Mo_2_C samples prepared under different holding times. The average grain size of the black hardcore phase in Figure 6a–d is 0.94 μm, 0.72 μm, 0.62 μm, and 0.75 μm.

When the holding time is 8 min, liquid-phase sintering is insufficient due to the short holding time, and the densification of the sample is incomplete, as illustrated in Figure 5a. At this time, the sample hybrid powder is not fully reacted, and the unreacted tungsten carbide and cobalt metal hinder the growth of Ti(C, N) grains, rendering the specimens fine-grained, highly porous, and resulting in a low density; this can also verify the reason for more WC being present as evinced by the previous XRD spectrum (Figure 1). As the holding time increases, the WC phase essentially disappears, solid-solution reactions become more complete and samples are gradually densified. After a holding time of 16 min (Figure 5c), the cermet sample shows the densest microstructures, with the fewest pores (both in number and size), and finer Ti(C, N) grains. Combined with Figure 1c, the diffraction peak of WC disappears, and the diffraction peak of Ti(C, N) shifts to higher angles at a holding time of 16 min. At this time, Mo and W form a stable rim phase (Ti, Mo, W) (C, N). This result is further demonstrated in Figure 5c, where a well-defined core-ring structure at the interface is observed, and a uniform distribution of W and Mo within the rim phase is evident from the EDS plot. Therefore, an appropriate holding time promotes the formation of the rim phase and inhibits the growth of core grains, leading to an improvement in the material properties. As the holding time is further prolonged to 20 min, as shown in Figure 5d, the material undergoes over-burning and pores appear therein. Therefore, the optimal holding time is 16 min at a sintering temperature of 1450 °C.

Figure 6 illustrates microstructures of the cermet sample with a molybdenum carbide content of 6 wt.% prepared at 1450 °C, holding for 16 min, under different sintering pressures. Under sintering pressure of 20 MPa (Figure 6a), relatively large grain sizes are found, and the average grain size is 0.81 μm for hard phases. At a sintering pressure of 25 MPa, a moderately thick grey annular phase of (Ti, M) (C, N) is formed around Ti(C, N) grains, inhibiting the growth of Ti(C, N) grains. At this point, black hard phases have a grain size of 0.61 μm and are more uniformly distributed. Under this pressure, there are almost no pores in the sample. Combined with Figure 7, Ti(C, N) cermet samples show optimal mechanical properties under a sintering pressure of 25 MPa. Optimal sintering pressure facilitates the formation of a uniform and dense core-ring structure. Conversely, excessive sintering pressure can result in abnormal growth of the black core phase. As the sintering pressure is increased, the hard phase exhibits abnormal growth in the grain size, reaching an average value of 0.87 μm at 35 MPa, and more pores appear in the sample. Combined with Figure 1d, the precipitation of the cobalt phase and graphite phase can be observed, indicating that excessive pressure can cause microstructural degradation.

### 2.3. Mechanical Properties

Mechanical property data of specimens obtained under different preparation conditions are displayed in Figure 7. As shown in Figure 7a, the densities and hardnesses of the samples change significantly with the change of Mo_2_C content under the condition of the constant sintering temperature of 1450 °C, holding time of 16 min, and sintering pressure of 30 MPa. Increasing the additive content shows a clear trend in which the density and hardness increase and then decrease.

The sample is found to have the optimal hardness, density, and fracture toughness at an Mo_2_C content of 6 wt.%. Compared with an Mo_2_C content of 0 wt.%, the hardness of the sample is increased by 39%, and the density is increased by 18%. The cermet specimen also has a high fracture toughness (8.3 MPa·m^1/2^), 51% higher than that of samples without Mo_2_C. The fracture toughness decreases by 12% when the content of molybdenum carbide is increased to 8 wt.%. This occurs because Mo_2_C, as an additive, not only modifies the wettability between the hard and bound phases, but also increases the thickness of the rim phase, which results in the densification of the samples, suppresses abnormal grain growth, and improves mechanical behaviors of the sample. According to previous reports, Mo_2_C improves the microstructure of the specimen, the microstructure of the alloy becomes finer, and the ring phase thickens with increased Mo addition [47,48]. With increasing Mo_2_C addition, the ring phase gradually thickens in the core-ring structure; because the grains are coarse and the brittle edge phase has a larger thickness, the composite tends to show deterioration of its mechanical properties as the Mo_2_C content is increased.

As shown in Figure 7b, the density of cermet samples increases with the sintering temperature, while their hardness shows an increasing then decreasing tendency. In Figure 7c, the values of both the density and hardness of specimens tend to increase and then decrease. When sintered at 1450 °C and with a holding time of 16 min, the samples achieve their highest hardness, density, and fracture toughness (HV 2845, 6.47 g/cm^3^, and 11.7 MPa·m^1/2^, respectively). These represent an improvement of 35% and 652% over those values at 1350 °C (4.78 g/cm^3^ and HV 378), and an improvement of 9% and 94% over those values at 8 min (5.93 g/cm^3^ and HV 1463). Furthermore, an increase of 102% in fracture toughness is observed for the samples compared to the value recorded under an 8-min holding time. However, this improvement is followed by a decrease of 34% in fracture toughness when the holding time is extended to 20 min. The mechanical properties of Ti(C, N)-based cermet samples may be degraded by excessive sintering temperatures and long holding times. This is mainly due to the fact that in the core-ring structure of Ti(C, N)-based cermet, the ring structure affects the properties of the samples. As mentioned above, the main component of the ring phase contains tungsten and molybdenum in the solid solution of (Ti, W, Mo)(C, N), which depends on the dissolution-precipitation of particles in the LPS process. Under the influence of appropriate sintering temperatures or holding times, the ring-phase thickness achieves an optimal balance, suppressing the growth of Ti(C, N) grains in the core-ring structure while facilitating fine-grain strengthening. On the other hand, if the temperature is excessively high or the holding time too long, particle dissolution-precipitation during LPS will be accelerated, and some small particles (carbide additives) will accelerate the precipitation and form a thicker ring structure with large particles; this over-thickened ring structure not only abnormally prevents grain growth, but also becomes more fragile, thus affecting overall mechanical behaviors of ceramic samples [49].

Mechanical behaviors of cermet specimens, containing 6 wt.% Mo_2_C and processed under ideal sintering conditions (1450 °C temperature and 16 min holding time), are displayed in Figure 7d for various pressure environments. Increasing the sintering pressure leads to a corresponding behavior in the hardness of the sintered sample, characterized by an initial increase followed by a subsequent decrease. At a specific sintering pressure of 25 MPa, the density of the sample stabilizes at 6.27 g/cm^3^. Concurrently, the hardness reaches its maximum value of HV 2731, while the fracture toughness is also maximized at 10.1 MPa·m^1/2^. In comparison with the hardness (HV 2360) at a sintering pressure of 20 MPa, the hardness value is increased by 15.70%, and the fracture toughness is increased by 8.60%. Compared with the hardness (HV 2558) at a sintering pressure of 35 MPa, its value is increased by 7% and the fracture toughness is increased by 11%. The density of the specimens varied relatively smoothly under conditions of varying pressure; however, the hardness tended to decrease under excessive pressure. With reference to Figure 1d, a possible reason for the decrease in hardness is that excessive sintering pressure causes the liquid phase (cobalt) to precipitate, which prevents the formation of a solid solution, resulting in the ring phase having an inhomogeneous thickness and the core structure exhibiting anomalous growth, ultimately affecting the mechanical characteristics of the ceramic samples.

Combining Figure 3 with Figure 6, it can be seen that the samples with smaller average grain sizes of the black core phase and a more uniform distribution of the core-ring structure exhibit better mechanical properties. This is because the unique core-ring structure significantly enhances the mechanical properties of the metal-ceramics. According to the Hall–Petch effect [49], a decrease in grain size leads to an increase in hardness. The complete and moderately thick annular phase effectively inhibits the growth of the black core phase, resulting in fine-grain strengthening and enhancement of the hardness of the samples. This also explains the higher Vickers hardness observed in the samples corresponding to Figure 3d, Figure 4c, Figure 5c and Figure 6b. The uniformly distributed rim phases (Ti, W, Mo) (C, N) contribute to high hardness and reduce the porosity of the samples [31]. As shown in Figure 7, the sample exhibits an optimal fracture toughness of 11.7 MPa·m^1/2^, with the corresponding SEM image provided in Figure 5c. From Figure 5c, it can be seen that the core-ring structure interface is well-defined, and the thickness of the ring phase is appropriate. As the crack propagates through the core-ring interface, it is deflected or blunted by the differences in interfacial mechanical properties. The deflection of the crack increases its propagation path, thereby dissipating more energy and enhancing fracture toughness. Additionally, the uniformly distributed bonding phase further contributes to the improvement of fracture toughness [16,17].

### 2.4. Magnetic Properties and Resistivity

Ti(C, N)-based cermets, with their better mechanical properties, have seen their popularity rise in industrial applications [50,51]. With the wide range of Ti(C, N) applications, the need for non-magnetic properties has also increased, as this is a prerequisite to machining ferromagnetic workpieces. The hysteresis loops displayed in Figure 8 portray the magnetic properties of the samples sintered via SPS under diverse conditions, providing experimental verification for the efficacy of Ti(C, N). As illustrated in Figure 8a, with molybdenum carbide as an additive to the process, the saturation magnetization strength (Ms) of the ceramic samples changed significantly. As the Mo_2_C content increases from 0% to 4%, the saturation magnetization strength of the samples decreases significantly from 10.1 emu/g to 8.4 emu/g. However, as the Mo_2_C content increases from 4% to 8%, the saturation magnetization strength of the samples increases from 8.4 emu/g to 10.2 emu/g. The magnetic properties of the ceramic samples are not only related to the binder phase (cobalt) but can also be influenced by the composition of the solid solution with elements such as tungsten, molybdenum, and titanium. Molybdenum and tungsten are non-magnetic elements. When combined with Figure 8a, it can be seen that a reasonable amount of Mo_2_C plays a part in suppressing the magnetic properties of the ceramic samples, which is attributed to the fact that the addition of the additive increases the molybdenum content in the binder phase (cobalt) on the one hand; moreover, on the other hand, it improves the wettability of the hard phase of Ti(C, N) with cobalt binder phase, enhancing the solid solution reaction of alloying elements (tungsten, molybdenum, and titanium).

Figure 8b shows that the saturation magnetization intensity of the samples increases with temperature. The saturation magnetization intensity is lowest at 1350 °C. As illustrated in Figure 4a, at lower temperatures, the liquid-phase sintering is incomplete, resulting in more pores and insufficient precipitation of magnetic particles (Co). This leads to the reduced continuity of the magnetic phase and a lower saturation magnetization intensity (5.9 emu/g). At 1450 °C, as seen in Figure 4c, the sample becomes nearly fully densified, the magnetic phase is fully precipitated, the magnetic particles are evenly distributed, and the saturation magnetization intensity reaches its maximum value (10.0 emu/g). Figure 8c shows the decreasing trend of magnetic behaviors of samples with the increase in the sintering time. Consistent with the XRD results in Figure 1, in time, the WC and Co phases disappear, resulting in the weakening of the magnetic properties of the sample. As illustrated in Figure 8d, the saturation magnetization intensity is the lowest and the magnetism is the weakest at the pressure of 35 MPa. Combined with the XRD analysis, it can be seen that when the pressure is 35 MPa, the sample exhibits the highest content of the Ti(C, N) phase, indicating lower solubility of WC and Mo_2_C in Co, resulting in lower magnetic properties. Meanwhile, the pressure of 25 MPa leads to a lower Ti(C, N) phase content, more sufficient solid solution reactions, and higher solubility of WC and Mo_2_C in Co, yielding stronger magnetic properties [52].

The resistivity of the specimens, produced under diverse conditions, is shown in Figure 9. Figure 9a,b shows that the resistivity of samples first decreases, then increases with the increasing molybdenum carbide content in the specimens and the rising sintering temperature. The sample in the third group of experiments (specimens with an Mo_2_C content of 0.4 wt.%) shows a minimum resistivity of 4.2 × 10^−5^ Ω·m. The resistivity of the specimens in the third group of experiments (specimens sintered at 1450 °C) shows a minimum value of 4.8 × 10^−5^ Ω·m. The electrical conductivity and electrical resistivity are inversely related; therefore, the addition of Mo_2_C promotes the increase in electrical conductivity in the samples. Mo_2_C is able to enhance the wettability of hard and binder (cobalt) phases, improve the ability of such metal ceramics to be sintered, and suppress the growth of Ti(C, N) particles to obtain metal ceramics with finer microstructures, better mechanical behaviors, and a certain degree of electrical conductivity; however, an excess of Mo_2_C in the cobalt will also limit the solubility of cobalt, resulting in consumption of more metal cobalt; therefore, given an excess of molybdenum carbide, the electrical conductivity of the sample will be decreased, and the electrical resistivity will be increased instead [43]. As shown in Figure 9b, if the reaction temperature is too low, the reaction is insufficient, and the resistivity of the sample is higher. When the reaction temperature is increased, the carbide is completely dissolved in the binder Co, and there is enough Co to dissolve the raw material at high temperatures; therefore, some of the Co remains, resulting in the lowest resistivity at 1450 °C and better conductivity. This is consistent with previous results, further evincing the maximum hardness and optimal densification after sintering at 1450 °C.

With reference to Figure 9c,d, it can be observed that the resistivity of the samples increases, then decreases, coinciding with both the extended sintering time and augmented sintering pressure. In the holding time group of experiments, the resistivity of the cermet sample in the second group of experiments shows a maximum value of 7.5 × 10^−5^ Ω·m. In the sintering pressure group of experiments, the resistivity of the sample in the second group of experiments shows a maximum value of 5.9 × 10^−5^ Ω·m. This is because, as the reaction time and sintering pressure increase, the solid-solution reaction is more complete, the content of metallic cobalt in the sample gradually decreases, the electrical conductivity decreases, and the resistivity increases.

## 3. Experiments

### 3.1. Preparation of Ti(C, N)-Based Cermet

Table 1 lists the key characteristics of the raw experimental powders. All the powders used in the experiments are supplied by Shanghai Shui Tian Material Science and Technology Co., Ltd. (Shanghai, China). The composition of the samples with different Mo_2_C contents is summarized in Table 2.

After weighing, the mixed powders were wet-milled in a planetary ball mill (QM-3SP2, Nanjing Leibu, Nanjing, China) using a ball-to-material ratio of 10:1, a rotational speed of 200 rpm, and anhydrous ethanol as the ball milling medium. The ball jar and balls were made of YG8 tungsten carbide. After the mixed powder was dried in an electric oven at 60 °C for 12 h, it was weighed (7 g) and placed in a graphite mold that had a height of 60 mm and an internal diameter of 20 mm. Between the powder and the mold was a layer of 0.2-mm graphite paper, interposed to facilitate the subsequent removal of the sample. The powder was compressed at room temperature using a powder compactor and then placed in an SPS furnace (SPS-30, Shanghai Chenxin, Shanghai, China). After setting the sintering pressure to reach 30 MPa, the program was initiated under vacuum (≤10 Pa) to elevate the temperature to 1450 °C, at which it was maintained for 16 min (content group, different Mo_2_C contents). At the conclusion of the holding period, the current was deactivated for the purpose of allowing the material to cool naturally. The sintering curves are depicted in Figure 10.

The application of a 2-min dwelling period at 600 °C, 800 °C, and 1000 °C enables the removal of oxygen from the powder mixture [53]. The sintered samples were cylindrical pieces that had a thickness of 3 mm and a diameter of 20 mm. In order to explore the effect of different molybdenum carbide contents on Ti(C, N), samples of the content group were first prepared. The raw material composition of the content group was (67-x)Ti(C, N)-25WC-8Co-xMo_2_C (x = 0.0 wt.%, 2.0 wt.%, 4.0 wt.%, 6.0 wt.%, 8.0 wt.%). Once tests of the content group were completed, various sintering temperatures (1350, 1400, 1450, and 1500 °C), sintering pressures (25, 30, 35, and 40 MPa), and holding times (4, 8, 12, and 16 min) were selected to facilitate the observation of the ways in which sintering conditions affect the structure and properties of the samples. The composition of the sample material used for the temperature, time, and pressure groups was 61Ti(C, N)-25WC-8Co-6Mo_2_C.

### 3.2. Characterization

The samples were ground and polished, followed by the measurement of their Vickers hardness and fracture toughness under 30 kgf using an HV-30Z Vickers hardness tester (Shanghai Jiezhun, China). Fracture toughness was calculated using Equation (5) [54,55]:*K_IC_ =* 0.0028 (*Hv* × *P/L*)^1/2^(5)
where *K_IC_* is the fracture toughness (MPa·m^1/2^), *Hv* stands for indentation hardness (kgf/mm^2^ or N/mm^2^), *L* represents the total crack length (mm), and *P* denotes the applied load (kgf or N).

An ED-300A densitometer (Shanghai Qunlong, Shanghai, China) was adopted to measure the density of each sample. A D8AA25 X-ray diffraction analyzer (Bruker, Berlin, Germany) was used to perform XRD analysis. After completing the aforementioned tests, the specimens were divided into 10 mm × 5 mm × 1 mm pieces using a wire cutter. The resulting sample pieces were then polished to facilitate further analysis.

Scanning electron microscopy (SEM) images were acquired using the back-scattered electron mode of a Sigma 300 field emission scanning electron microscope. Elemental distribution on the sample surface was analyzed using mapping. XPS spectra were acquired using an XSAM800 photoelectron spectrometer (Kratos, Manchester, UK). To ascertain the magnetic properties and resistivity of samples under examination, both the JDAW-2000D vibrating sample magnetometer and the ST2253 multi-function digital four-probe tester were utilized for testing purposes.

## 4. Conclusions

A spark-discharge plasma sintering system was employed for producing Ti(C, N)-based cermets containing Mo_2_C and tungsten carbide under various process conditions. The key conclusions arising from the experimental work are as follows:When keeping the sintering temperature, holding time, and sintering pressure constant at 1450 °C, 16 min, and 30 MPa, the addition of Mo_2_C shifts the XRD diffraction peaks of Ti(C, N) to lower angles, while a more complete core-ring structure emerges in the microstructure of Ti(C, N). This phenomenon induces alterations in the mechanical properties of the cermet samples. With a Mo_2_C addition of 6 wt.%, the sample exhibits the smallest grain size of the black hard phase (0.64 μm) and a well-distributed core-ring structure. The density of the sample is 6.45 g/cm^3^, the Vickers hardness is HV 2318, the fracture toughness is 8.3 MPa·m^1/2^, the saturated magnetization strength is 11.4 emu/g, and the resistivity is 5.2 × 10^−5^ Ω·m;With a Mo_2_C addition of 6 wt.%, keeping the same sintering pressure (30 MPa) and holding time (16 min) while varying the sintering temperature, the XRD diffraction peak intensities of Ti(C, N) and WC decrease. At a sintering temperature of 1450 °C, the Ti(C, N) grains achieve their smallest size (0.58 μm), accompanied by a uniform thickness of the rim phase and optimal mechanical properties of the sample. Under these conditions, the sample exhibits a density of 5.96 g/cm^3^, a Vickers hardness of HV 2318, a fracture toughness of 8.2 MPa·m^1/2^, a saturation magnetization strength of 10.0 emu/g, and a resistivity of 4.8 × 10^−5^ Ω·m;With a Mo_2_C addition of 6 wt.%, maintaining a sintering temperature of 1450 °C and a sintering pressure of 25 MPa, and varying the holding time, the XRD diffraction peaks of Ti(C, N) shift to lower angles as the holding time increases. At a holding time of 16 min, the sample exhibits a clear core-ring interface, a uniformly thick ring phase, and the smallest hard phase grain size (0.62 µm). Under these conditions, the sample achieves a density of 6.47 g/cm^3^, a Vickers hardness of HV 2845, a fracture toughness of 11.7 MPa·m^1/2^ a saturation magnetization strength of 7.25 emu/g, and a resistivity of 5.3 × 10^−5^ Ω·m;With a Mo_2_C addition of 6 wt.%, the optimal sintering conditions of the samples are a sintering temperature of 1450 °C, a holding time of 16 min, and a sintering pressure of 25 MPa. Under these conditions, the prepared samples exhibit the best properties, including a density of 6.27 g/cm^3^, Vickers hardness of HV 2731, fracture toughness of 10.1 MPa·m^1/3^, saturated magnetization strength of 10.3 emu/g, and resistivity of 5.5 × 10^−5^ Ω·m. In comparison to samples without Mo_2_C, the presence of Mo_2_C resulted in notable improvements in hardness, density, and fracture toughness, with increases of 63%, 15%, and 84%, respectively.

## Figures and Tables

**Figure 1 molecules-30-00492-f001:**
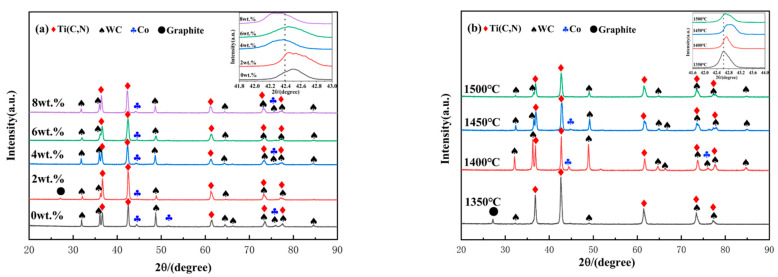
XRD diagrams of Ti(C, N)-based cermets sintered using SPS under different conditions: (**a**) the Mo_2_C content; (**b**) sintering temperature; (**c**) holding time; (**d**) sintering pressure.

**Figure 2 molecules-30-00492-f002:**
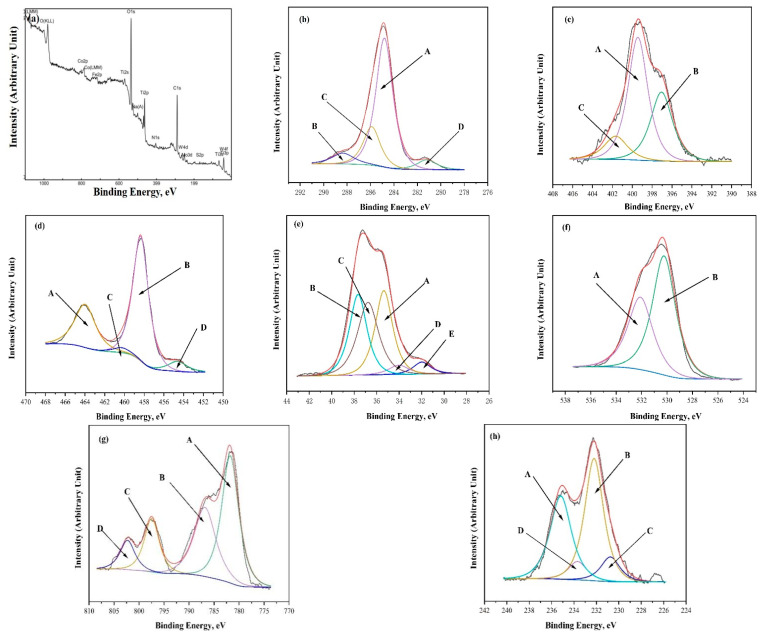
XPS spectra of the sample (Ti(C, N)-WC-Co-Mo_2_C) obtained at 1450 °C, 16 min, and 30 MPa: (**a**) full spectrum; (**b**) C1s; (**c**) N1s; (**d**) Ti2p; (**e**) W2p; (**f**) O1s; (**g**) Co2p; (**h**) Mo3d.

**Figure 3 molecules-30-00492-f003:**
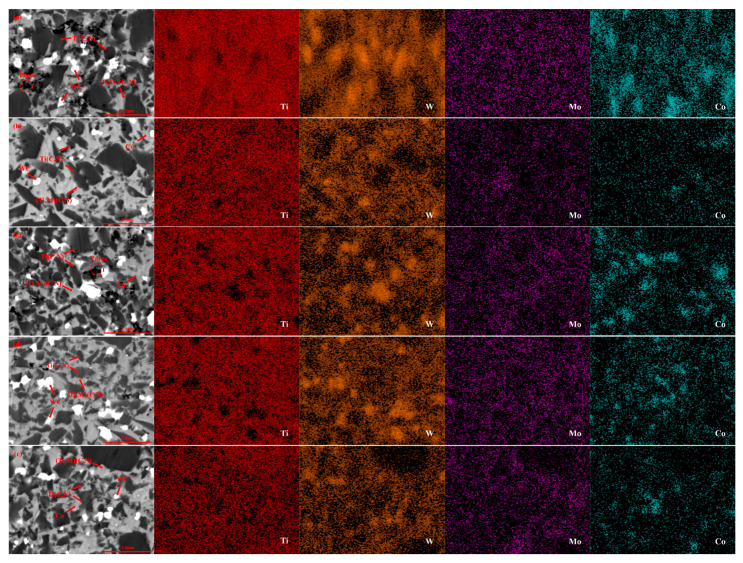
Backscattering and mapping diagram of Ti(C, N)-based cermets with different contents of Mo_2_C: (**a**): 0 wt.%; (**b**): 2 wt.%; (**c**): 4 wt.%; (**d**): 6 wt.%; (**e**): 8 wt.%.

**Figure 4 molecules-30-00492-f004:**
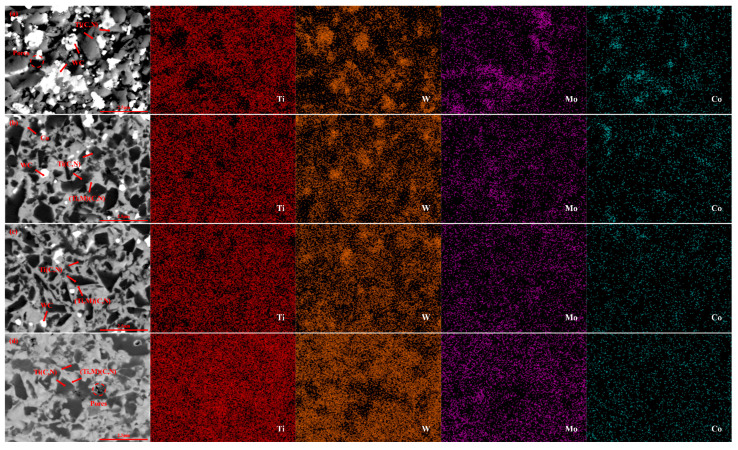
Backscattering and mapping diagram of Ti(C, N)-based cermets at different sintering temperatures (**a**): 1350 °C; (**b**): 1400 °C; (**c**): 1450 °C; (**d**): 1500 °C.

**Figure 5 molecules-30-00492-f005:**
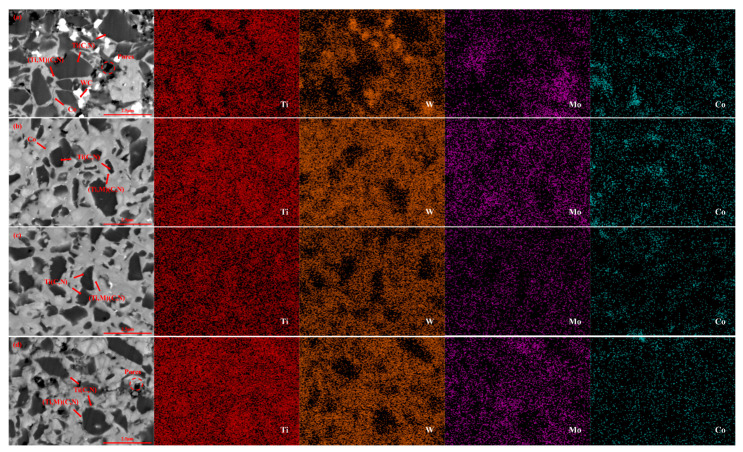
Backscattering and mapping diagram of Ti(C, N)-based cermets at different holding times: (**a**): 8 min; (**b**): 12 min; (**c**): 16 min; (**d**): 20 min.

**Figure 6 molecules-30-00492-f006:**
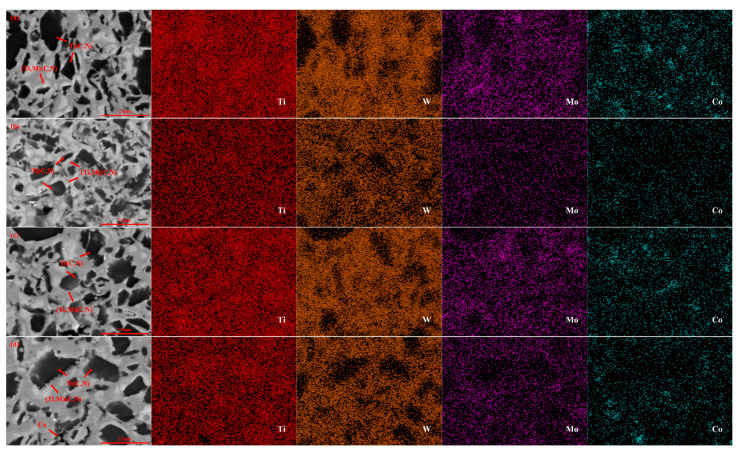
Backscattering and mapping diagram of Ti(C, N)-based cermets at different sintering pressure: (**a**): 20 MPa; (**b**): 25 MPa; (**c**): 30 MPa; (**d**): 35 MPa.

**Figure 7 molecules-30-00492-f007:**
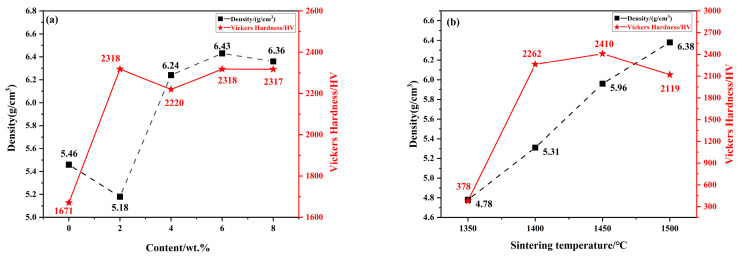
Mechanical properties of Ti(C, N)-based cermets prepared using SPS under different conditions: (**a**) different Mo_2_C contents; (**b**) different sintering temperatures; (**c**) different holding times; (**d**) different sintering pressures; (**e**) different samples.

**Figure 8 molecules-30-00492-f008:**
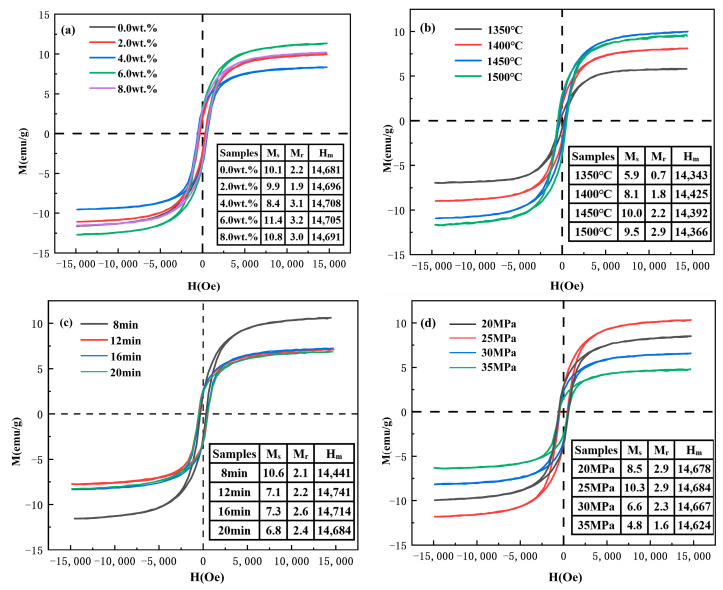
Hysteresis loops of Ti(C, N)-based cermets prepared using SPS under different conditions: (**a**) different Mo_2_C contents; (**b**) different sintering temperatures; (**c**) different holding times; (**d**) different sintering pressures.

**Figure 9 molecules-30-00492-f009:**
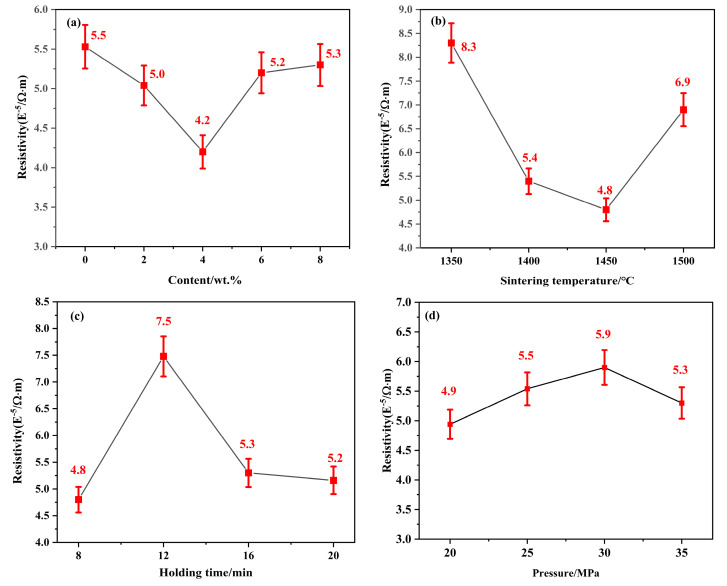
Resistivity of Ti(C, N)-based cermets prepared using SPS under different conditions: (**a**) different Mo_2_C contents; (**b**) different sintering temperatures; (**c**) different holding times; (**d**) different sintering pressures.

**Figure 10 molecules-30-00492-f010:**
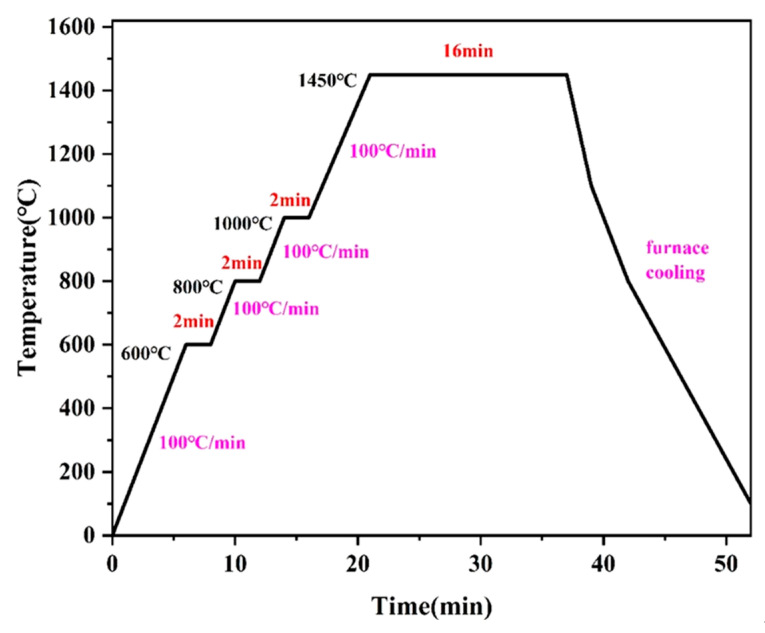
Schedule of SPS.

**Table 1 molecules-30-00492-t001:** Characteristics of the raw powders.

Powder	Ti(C, N)	WC	Co	Mo_2_C
Particle size (μm)	0.62	0.2	0.05	3
Purity	99.9%	99.9%	99.9%	99.9%

**Table 2 molecules-30-00492-t002:** Composition of the samples.

Ti(C, N) (wt.%)	WC (wt.%)	Co (wt.%)	Mo_2_C (wt.%)
67	25	8	0.0
65	25	8	2.0
63	25	8	4.0
61	25	8	6.0
59	25	8	8.0

## Data Availability

Data will be made available on request.

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
