# Peer review of "Effects of Mo2C on Microstructures and Comprehensive Properties of Ti(C, N)-Based Cermets Prepared Using Spark Plasma Sintering"

_molecules, 2025, doi:10.3390/molecules30030492_

Round 1

Reviewer 1 Report

Comments and Suggestions for Authors

-          Abstract: please correct the unit of density (g/cm³)

-          Section 2.1: “the initial powder mixture was wet milled […], followed by mixing in a planetary ball mill“. How was the powder mixture wet milled? Do you actually mean that the powder mixture was mixed and milled in the planetary ball mill (which is the usual processing)?

-          Please clarify what the “content group“ is. Please clarify “Once tests of the content group were completed, […]” – which kind of tests were completed before the variation of sintering temperature and time? This is not clear at all in the beginning and only begins to make sense after finishing to read the whole paper.

-          Fig 2b,c  and d: please specify the composition of the samples.

-          Fig 4: please specify sintering conditions

-          Fig 5 and 6: please specify sample composition

-          Bright field and dark field are different imaging techniques. WC is the bright phase and Ti(C,N) the dark phase.

-          What do you mean by “SEM-BSE _model_”? Using a BSE detector of a SEM results in images where heavy elements appear brighter than lighter ones.

-          Overall the correlation between the Backscattering _image_ and the EDS mappings seem to be pretty poor due to EDS mapping acquisition parameters. Usually the microstructure/grains are reflected in the EDS mappings.

-          “holding time is 8 minutes, sintering has stopped due to the short time”: sintering / densification by sintering is not completed.

-          Please specify how the grain size was measured. There are different techniques and also different kind of “grain sizes” (intercept length, Feret diameter, ECD..)

-          “combined with Fig 7…” should this be Fig 8 (figure with mechanical data?)

-          Fig 8: typo in legend and axis label: “Vickers hardness”

-          Fig 8: please round the values of Vickers hardness to reasonable values (2318 HV instead of 2318.23) in regard to measurement uncertainty

-          In my opinion also the percentages of increasing/decreasing properties should be rounded and two decimal points is too much/makes it hard to read.

-          Fig 8 typo in bar diagram: legend: “content”

-          Fig 8 and 9 and 10: again, which composition was used to test different sintering conditions? This info is missing in the Experimental section

-          Section 3.3: please clarify what you mean by “cyclic phase”. Is this the rim phase?

-           Mo2c addition and influence on magnetic saturation: “Ms.. has been significantly reduced”: This is not clear. Do you mean the decrease from 10.12 (0% Mo2C)to 8.36 (4 % Mo2C)?

-          Solubility usually increases with increasing temperature. Are you sure that the solubility of WC/Mo2C in Co is higher at 1350 °C compared to 1450 °C?

Comments on the Quality of English Language

The quality of English is acceptable, but a bit hard to read. If you have the possibility to get feedback from someone more proficient- please revise

Reviewer 2 Report

Comments and Suggestions for Authors

The submitted manuscript is very interesting. The authors did a lot of research and checked the influence of sintering time, sintering pressure on material properties and structure. 

However, minor comments should be corrected before publication: 

1. The SEM microscope images are very small and therefore not readable. They should be improved. In addition, the markings on them are also not visible, maybe change the font colour? 

2. In my opinion, the conclusions are too short. I think with such extensive studies, it is worth adding more conclusions. 

Reviewer 3 Report

Comments and Suggestions for Authors

Revision of manuscript no. molecules-3348308

This paper reports an investigation into the effects of Mo2C on the microstructures and comprehensive properties of Ti(C, N)-based cermets prepared using spark plasma sintering (SPS). Overall, the manuscript is well-structured, and the research is thoughtfully designed. The topic is both scientifically and technologically significant, with the potential to interest the scientific community in the field of advanced ceramic materials and manufacturing. However, in my opinion, some enhancements and clarifications are needed before publication in Molecules. In this regard, I recommend a Major Revision.

Comments/Questions:

1.      Could the authors clarify how the core-ring structure in Ti(C,N)-based cermets improves their performance in practical applications? Does its presence directly improve wear resistance, toughness or some other application-relevant property of the cermet?

2.      Were control samples or reference standards used to validate mechanical test results for hardness and toughness?

3.      Did the XRD analysis use Rietveld refinement or any other quantitative phase analysis method to determine the proportion of phases present in the sintered samples?

4.      Were Vickers hardness measurements taken on multiple areas of the same specimen to ensure uniformity?

5.      What method was used to calculate the fracture toughness? Please give the formulae used and the relevant experimental conditions.

6.      How many specimens were tested for each test condition in the hardness and fracture toughness tests? How were the mean values and standard deviations calculated?

7.      In the SEM analysis how the grain size distribution and ring structure formation were quantified? Were any image analysis software or quantitative methods used to measure grain size and ring thickness?

8.      Could you explain the mechanism behind the development of the core-ring structure in Ti(C, N)-based cermets? How do sintering conditions (temperature, pressure and holding time) affect nucleation and growth of core and rim phases, and how do diffusion and dissolution-precipitation mechanisms affect grain growth and ring formation?

9.      How does the formation of the ring structure affect the mechanical properties of the cermet, particularly in terms of hardness and fracture toughness? Could you clarify if the presence of the ring structure contributes to grain refinement or to improved bonding between phases?

10.  In Figure 10, authors must standardise the number of significant figures for each value reported in the graph and include error values associated with each measurement.

11.  The conclusions section omits the results of SEM, XRD, XPS, magnetic and electrical properties. Why is there no reference to all the results of the work?

12.  How can the obtained results be applied in industrial processes for manufacturing cermets? Are there any limitations to implementing the described experimental conditions in a real production environment?

Round 2

Reviewer 3 Report

Comments and Suggestions for Authors

Revision of manuscript no. molecules-3348308_Round#2

I thank the authors for their efforts in addressing the previous comments and for significantly improving the manuscript. However, there is still one important aspect of the fracture toughness calculation that needs clarification. Therefore, I recommend a Minor Revision before the article can be accepted for publication.

Comments/Questions:

1.      Clarification of fracture toughness formula: please provide the source or reference for the formula used to calculate fracture toughness. The current formula raises some concerns about its validity. I strongly recommend that the authors consult the ASTM C1327-15 standard and refer to the method proposed by Anstis et al. (1981) for Vickers indentation fracture toughness calculations: https://doi.org/10.1007/978-3-319-92955-2_4. This will help to ensure the accuracy and scientific validity of the results.

2.      After reviewing and correcting the fracture toughness calculation, if necessary, the authors should expand the analysis in the discussion, particularly in response to question 9 from the first review. A more detailed explanation of how the corrected toughness values relate to microstructural features (e.g. core ring structure) and mechanical performance would strengthen the manuscript.
